# Towards a More Accessible Cultural Heritage: Challenges and Opportunities in Contextualisation Using 3D Sound Narratives

**Veranika Lim** [1,*,†], **Sara Khan** [2,†] **and Lorenzo Picinali** [1,*]

1   Dyson School of Design Engineering, Imperial College London, London SW7 2DB, UK
2   Department of Architecture and Design, Politecnico di Torino, 10129 Torino, Italy; khan.saretta@gmail.com
*   Correspondence: v.lim@imperial.ac.uk (V.L.); l.picinali@imperial.ac.uk (L.P.)
†   These authors contributed equally to this work.

**Abstract:** This paper reports on the exploration of potential design opportunities for social media and technology to identify issues and challenges in involving people in generating content within a cultural heritage context. The work is divided into two parts. In the first part, arguments are informed by findings from 22 in-depth semi-structured interviews with representatives of cultural institutions and with people from a general audience who recently participated in a cultural activity. The key findings show that social media could be used more extensively to achieve a deeper understanding of cultural diversity, with opportunities in *redefining the expert*, *extending the experience space*, and *decentralising collaboration*. To further support these findings, a case study was set up evaluating the experience of a mini audio tour with user-generated (i.e., personal stories from a local audience) vs. non user-generated (i.e., professional stories including facts) narratives. These were delivered using text and 3D sound on a mobile device. The narratives were related to a built environment in central London near world-renown museums, cultural buildings, and a royal park. Observations, a standardised spatial presence questionnaire, and a short open interview at the end of the tour were used to gain insights about participants preferences and overall experience. Thematic analysis and triangulation were used as a means for understanding and articulating opportunities for social media to better involve and engage people using user-generated narratives presented through 3D sound.

**Keywords:** cultural heritage; sound and narrative design; built environment; user-generated content

## 1. Introduction

Living in the era of an always-connected society has created great opportunities for developing technological tools to support distributed curation in cultural heritage. The most known "place" for distributed curation is social networks, also called user-generated content (UGC). In the last decade, an increasing number of cultural heritage venues have started to adopt the use of social networks to facilitate cultural contributions from everyday visitors [1]. As a result, there is an continuously increasing amount of UGC on the Web.

Despite the "explosion" of UGC, only a small portion of people contribute to generating new content. There is an emerging rule of thumb that suggests that, if you get a group of 100 people online, then one will create content, 10 will "interact" with it (commenting or offering improvements), and the other 89 will just view it [2]. Furthermore, although social media is currently being used by many cultural institutions to reach their audiences, our findings show that there is one mayor issue; the large amount of information provided and available online is problematic for an efficient reach to potential audiences. To our knowledge, however, distributed curation has not been fully exploited with regard to cultural heritage promotion and integration in people's everyday lives.

This paper reports on an exploration of potential design opportunities for social media and technology to identify issues and challenges in involving people in distributed curation through 2 studies. In the first study, we took an empirical and critical approach using data from in-depth, semi-structured, and audio-recorded interviews to identify potential

design opportunities. In the second study, we aimed to support the interview results using a case study. In the case study, we explored how user-generated content could contribute to visitor's engagement around built heritage in an "everyday life" context using 3D sound narratives. Overall, this work is conducted as part of a larger project called *PLUGGY* [3], which aims to allow users to share their local knowledge and everyday experience with others, and alongside cultural institutions. PLUGGY aims to build extensive networks around a common interest around cultural heritage.

In the following sections, we first introduce an emerging paradigm shift to contextualise our approach. We show a number of recent examples of projects in Human–Computer Interaction (HCI) that aimed to support distributed curation using social media. We then present the methods and outcomes of a series of in-depth semi-structured interviews. Themes are discussed on how existing social media could be better utilised to improve user engagement and to encourage distributed curation within a cultural heritage context. We then present the case study where we evaluated the experience of a mini audio tour with user-generated (i.e., personal stories from a local audience) vs. non user-generated (i.e., professional stories including facts) narratives.

## 2. Background

### 2.1. A New Paradigm in Cultural Heritage

In the traditional view of cultural heritage, people valued material things with intrinsic properties or a national history relevance. The traditional paradigm encouraged the reduction of heritage to tourism and consumption, with the engagement of broader cultural topics only for a professional audience. Recently, however, a new paradigm emerged that puts the production of heritage to the foreground. The new paradigm aims to encompass the greater involvement of local citizens in creating content and to contribute to cultural heritage, with greater concern for the everyday. Generally, it has been argued that, when heritage is everywhere and relevant to our everyday life, it is likely to be one of the preconditions for genuine sustainability [4]. This stresses the importance of heritage communities, which are social groups, who value specific aspects of cultural heritage that they wish to sustain and transmit to future generations within the framework of public action [5] and social capital, which is defined as an important resource of individuals and social groups impacting economic growth, democratic practices, quality of governance, and quality of life [6]. Hence, local citizen participation is considered an ethical obligation and a political necessity to strengthen democracy and to create governance that can ultimately improve quality of life [4].

Some key examples of enablers of the paradigm shift are *policy*, *inclusivity*, and *technology*. Firstly, the role of culture as a component of sustainable development is being increasingly discussed in policy debates. UNESCO emphasised the importance of culture during the Decade of Culture and Development (1988–1998) and through its conventions (e.g., on the Protection and Promotion of the Diversity of Cultural Expressions in 2005, for the Safeguarding of the Intangible Cultural Heritage in 2003, and concerning the Protection of the World Cultural and Natural Heritage in 1972). In Europe, both the Council of Europe's Landscape Convention and its Faro Convention imply the need for more culturally sensitive approaches [4].

Secondly, heritage is often represented by places linked to a built environment that is already iconic but not really representative of all the sides and transliterations of a culturally diverse audience [7]. This happens also on a smaller scale in museum environments, e.g., the Mona Lisa in the Louvre museum is one of the most famous masterpieces, but most of the time, people focus just on this artwork and easily forget to pay attention to the surroundings. According to a study published by the MIT Senseable Lab [8], generally, visitors overlook the very famous Giotto's display just in front of da Vinci's masterpiece. The reason for this is because anyone who visits the museum feels the need to snap a picture of the Mona Lisa, as a means to underline the *"Louvre museum experience"* and *"to bring back home the proof of having been there"* as a small souvenir. Therefore, collecting samples of

the museum/cultural experience is a recurring action for tourists and museum visitors [9], while cultural landmarks are becoming similar to brands. To reach better visibility, cultural heritage needs to be understood by a wider range of visitors. One key aspect in being understood by the audience is to find a common root between different cultures and to globalise what, at the moment, is only locally known [9]. The challenge here is to contextualise masterpieces, cultural landmarks, and pieces of the built environment to a culturally diverse panorama. Cultural artefacts are rarely located where most people live or work: *here*. Too often they are—it seems almost by definition—somewhere else: *there*. People might visit them on holiday, but this type of heritage is not part of the everyday landscape of their normal lives. If heritage is thus defined as elsewhere, there is a risk that it will unintentionally become an instrument of exclusion. It can only become an instrument of inclusion and commonality if it is defined contextually as local, lived-in, and ordinary and if it is recognised as an element of both shared identity and differentiation [10]. According to Giaccardi, contemporary heritage studies are the results of ongoing interactions in the lived world of ordinary people [11].

Thirdly, because of careful planning of mobile technology manufacturers and telecommunications service providers, we have now reached an always-connected society [12]. The combination of ubiquitous access to multimedia content and information with the consolidation of distributed curation generates great opportunities to develop technological tools to further enable this paradigm shift in cultural heritage.

### 2.2. Distributed Curation in Social Media and HCI

Distributed curation is a phenomenon by which citizen communities have accepted freely providing content and information [13]. Digital and social technologies are facilitating distributed forms of curatorial practice, which can be harnessed to democratise history [14], in turn supporting inclusivity and policy. Liu [14] describes distributed curation as a sociotechnical practice involving people, cultural artefacts, and information and communication technology. It is a collaborative and distributed practice, creating shared ownership over the stewardship of the living heritage through transparency, which further allows other parties to partake in the curatorial process.

There have been a number of recent projects in HCI that aimed to support distributed curation using social media. One well-known social media channel generally accepted and used by communities is Instagram. Instagram allows creative practices from non-elite social contexts and communication that relies on everyday competencies rather than on formal artistic education (Burgess, 2006 as described in [15]), extending the reach of the museum beyond its walls. An example is the *CURIOS* project [1]. *CURIOS* researchers explored how digital archives for rural community heritage groups can be made more sustainable so that volunteer members can maintain a lasting digital presence. They developed software tools to help remote rural communities collaboratively maintain and present information about their cultural heritage using open linked data. This approach is adjusted to the specificity of a local heritage but can also take advantage of already collected materials from elsewhere. In another example, Yelmi et al. [16] evaluated soundscapes as an intangible cultural heritage element and introduced the *Soundsslike* project. *Soundsslike* is a crowd sourced online sound archive that invites people to record symbolic urban sounds and to upload them to an online sound archive. This online platform was built and displayed in an exhibition through an interactive table-top interface to learn more from users and contributors and to enrich the archived content by raising public awareness of urban sounds.

Distributed curation has also been shown to be helpful in personalising experiences. For example, *MobiTag* [17] is an electronic guide that supports semantic, social, and spatial navigation in museums by allowing visitors to create and vote for tags. Han et al. [18] developed a mobile application called *Lost State College* (LSC) and showed that users used social features as a way of learning local history and of interacting with others, co-creating digital traces and rich layers of local history information. McGookin and Brewster designed

*PULSE* [19], which allows users to gain a vibe (i.e., an intrinsic understanding of the people, places, and activities around their current location) using Twitter data. As users moved, *PULSE* downloaded public messages (tweets) generated by any user in the current location. Then, *PULSE* selected the closest tweet and inserted it in a virtual 3D auditory environment: users heard tweets as whispered conversations. Bellens et al. [20] explored how social media data can be employed to study tourism on European cultural routes and showed its potential for investigating a complex touristic object such as a cultural route. They combined text related to photos on Instagram with Wikipedia for geographical places. This allowed them to identify the most popular stops and localities related to the cultural route. In addition, Bujari et al. [21] proposed *PhotoTrip*, an interactive tool able to autonomously recommend cultural heritage locations along travel itineraries even if those locations were not mainstream. *PhotoTrip* identified points of interest by gathering pictures and related information from Flickr and Wikipedia and then by providing the user with suggestions and recommendations.

Through these examples, we can see that communities are being more and more involved through distributed curation, where technologies mediate and allow users to contribute to our histories. This supports what was found in previous work, claiming that awareness towards local heritage should be promoted at first among locals in their living or working environment [22–24]. The results of these projects and of the current use of social platforms have demonstrated their potential to build networks through the individual and distributed contributions of users. To our knowledge, however, these possibilities have not been fully exploited with regard to cultural heritage promotion and integration in people's everyday life. We believe that distributed curation is still limited in the cultural heritage sector. Most of the time, tourists deal with specific tourism-related agents and locations, such as tour guides, desk clerks, and taxi drivers, and rarely merge with the surrounding community [25]. Mass tourism can create a barrier between visitors and locals. It is therefore not surprising that a wide range of new concepts such as "sustainable tourism", "resident responsive tourism", and "community-based tourism" have become important. Visitors are in constant need of help in finding relevant information, but providing them with appropriate information is challenging because their interests and needs are unknown [26].

This paper reports on two studies we conducted to explore potential design opportunities for social media and technology and to identify issues and challenges in empowering people to participate in cultural heritage curation. The first study took an empirical and critical approach using data from in-depth, semi-structured, and audio-recorded interviews. In the second study, we aimed to support the results coming from the interviews through a case study. For the case study, we looked at broadening the narrative scope of a built environment using curator vs. user-generated 3D audio narratives. The aim was to explore how user-generated content could contribute to visitor's engagement within an "everyday life" context, and to better understand the challenges we face and the opportunities we can utilise to enhance engagement.

## 3. Study 1: In-Depth Interviews

In-depth interviews are the cornerstone of design thinking. Through research techniques such as in-depth interviews, we can learn how different users or customers feel about the problem one is trying to solve and how they might fix it if they could.

### 3.1. Methodology

#### 3.1.1. Participants

We conducted in-depth semi-structured interviews with 22 participants who could potentially be early adopters of PLUGGY. The participants were divided into two groups: cultural heritage professionals and a general audience who recently participated in cultural activities. Heritage professionals were recruited through our immediate social networks and referrals working in the cultural heritage industry. For the general audience target,

we recruited connections within our social network who have recently visited a cultural heritage site. Interviewing two different stakeholder groups allowed us to see the different perspectives in motivations, challenges, and design opportunities. Cultural heritage professionals varied in occupation including museums' head of media, curators, guides, administrators, and managers; an architect; an economist; and a journalist. All worked in either museums, galleries, or independently, but all within the field of cultural heritage. The general audience participants included post-graduate students, researchers, a developer, a company founder, an account assistant, and a manager, who all often engage in and have recently engaged in leisure cultural heritage activities.

### 3.1.2. Procedures

Cultural heritage professionals were asked questions covering themes such as their background, recent projects and their motivations and goals, their social media and social technology use, and the main challenges in these. Other key themes were related to their views of what the future should bring in terms of experiences and the role of social technology and social media. General audience participants were asked questions covering themes such as personal background, their recent cultural heritage-related activities, how they use social media in their activities, their motivation and goals, and what challenges they commonly face. See Appendix A for the complete list of questions. The interviews lasted 30 to 90 min in Q3 of 2018 and took place in participants' respective locations or online using Skype. Interviews were audio recorded for transcription purposes.

### 3.1.3. Analysis

We conducted a thematic analysis in which we analysed the interviews in steps, gradually identifying emerging themes in the data. After the interviews were transcribed verbatim, items (i.e., paragraphs) were selected when relevant to our goals (i.e., highlighting opportunities to engage people better in the cultural heritage sector and exploring how current social media could further be utilised). Relevant items were then coded individually and discussed collaboratively by two authors. The MAXQDA (https://www.maxqda.com/, accessed on 26 March 2021) qualitative analysis software was used to map the frequency and relevance of codes. Based on these, we identified 3 main overarching themes, which are discussed in the next section using pseudonyms to refer to participants.

### *3.2. Results*

We collected and transcribed a total of 930 min of audio-recorded interviews (approximately 27,500 words). A total of 283 comments were identified (one comment can be a sentence or a paragraph describing, for example, an issue, reflection, or concern); 165 comments were categorised to fit within 3 overarching themes selected for discussion in this paper. Comments that were mentioned only once were not considered as empirically relevant, and they were therefore left out of the analysis.

Overall, we identified three overarching themes, each identifying opportunities or implications for further exploration. These three themes are *redefining the expert*, *extending the experience space*, and *decentralising collaboration*. For each theme, we discuss the participants' main challenges, with an attempt to identify potential design opportunities for social media to promote cultural heritage and to foster cultural diversity.

### 3.2.1. Theme 1: Redefining the Expert

Firstly, our findings suggest considering local audiences as experts and, therefore, as an important resource and potential marketing channel. This adds to Kidd's argument about the relevance of promoting the "face" of an institution through local visitors or local residences, who should play a more significant role in creating an institution's face in promoting cultural heritage [27].

The following findings are also in support of this argument. Both cultural heritage professionals and general audience interviewees shared challenges related to finding

content and events (related to 23 and 39 comments, respectively). Participants felt biased in discovering culture-related content as well as no sense of fulfillment in finding content at all. They mentioned being aware of the results of the algorithms to dictate what we encounter online. This, also called *filter bubble*, represents a state of intellectual isolation that can result from personalised searches when a website algorithm selectively guesses what information a user would like to see based on information about the user, such as location, past click behaviour, and search history [28]. General audience interviewees, for example, expressed the feeling of being overwhelmed or overloaded by the events, activities, and content promoted on social media pages such as Facebook. Specifically, events and content related to dance culture and folklore (i.e., intangible heritage) are mentioned as an even bigger challenge for discovery. This is understandable as Bea, a cultural heritage professional we interviewed, explains that *"almost every cultural institute, if not all, are reaching their audiences using similar standard means such as Facebook, Twitter, and Instagram"*. Likewise, Fabio, who is a journalist in Greece, seemed to have difficulties in reaching out to potential audiences in other countries such as Germany because he believes that the more audiences he has in Greece, the more his content is automatically shown to potential audiences in Greece.

In response to these challenges, general audience interviewees rely strongly on information from their immediate friends or from locals nearby during their cultural visits, as they have low confidence and trust in social media content for the reasons described above (related to 18 comments). These kinds of mouth-to-mouth discoveries usually take place by direct messaging, face to face, or when friends take one another on a cultural visit. Interviewees expressed their preference in approaching locals or connections as they hold immediate and trustful expertise of information beyond what is publicly promoted.

Examples of related work focusing on local contributions to cultural heritage can be found in the application developed by Han and colleagues [18] called *Lost State College* (LSC). This research shows that users employed social features as a way of learning local history and interacting with others, co-creating digital traces and rich layers of local history information. Another example is the *CURIOS* project [1], where researchers explored how digital archives for rural community heritage groups can be made more sustainable so that volunteer members can maintain a lasting digital presence. Something similar but for sound was introduced by Yelmi and colleagues [16]. Another platform named *ArtLinks* was developed to provide a guidance system based on a public display in museum exhibits, which allowed visitors to create and use geo-tags to help guide other visitors [21]. Similarly, *MobiTag* [17] is an electronic guide that supports semantic, social, and spatial navigation in museums by allowing visitors to create and vote for semantic tags to describe objects.

Although these examples are headed towards the right direction in redefining an archetypal expert, most of these studies were done within the conventional museum space. We believe that current social media and technologies do not sufficiently support local audiences in taking the role of the "expert", utilising their knowledge significantly in the everyday space. In order to do this, we need to extend the experience space beyond museums, archives, or specific heritage sites and allow local audiences to contribute and share heritage-related content easily.

### 3.2.2. Theme 2: Extending the Experience Space

Secondly, our findings show an opportunity in blending discovery and experience, e.g., what happens at home or in a hotel room could transition to what happens on site. Dialogue in social media often comes from a small number of contributors or active community members [29]. To encourage and enable more genuine participation, dialogue, and inclusive spaces, content could be made more dynamic and interesting for a broader audience who are navigating in a broader space. Kidd [27] discussed two examples of good practice for inclusive spaces, highlighting the need to accommodate different audiences. Examples of institutions that went in this direction are the Powerhouse Museum in Sydney, which has relaxed its policy, recognising that taking photographs is an important part of the experience for visitors. On a similar pathway, the Australian Museum started making

a distinction between the needs of different kinds of "visitors" by hosting separate blogs. When aiming at further promoting inclusive spaces beyond museum spaces, our findings show an opportunity in blending discovery and experience.

Interviewees usually plan high-level activities and allow space for discoveries on site and in the moment. They tend to use cues within their environment (e.g., large colourful objects or crowds) to lead their attention and to make impromptu decisions. This is a great thing according to our professional interviewees, as it allows for serendipity. However, serendipity may decrease visitors' motivation to download an application that can be used in an exhibition or museum to enforce engagement. There are challenges in asking visitors to use their devices. They may face barriers such as limited internet access, limited time in understanding the app's features, or not enough memory space to download yet another app they may not use again after their visit. According to our interviewee Alfred, who is responsible for media, visitors do not want to download something for a visit of a couple of hours and then never use it again. Consequently, visitors may just come and see beautiful objects with limited engagement. Because of this, museums or cultural spaces are not perceived as part of visitors' everyday space. Technology limitations may making it difficult to transition between spaces because applications that are used on an everyday basis are not connected to museum or cultural spaces whatsoever.

Cultural heritage professional interviewees have focused too much on experiences inside a cultural institution or site as a way to improve visitor engagements with objects. However, they acknowledge the need to better connect objects with audiences' personal everyday lives off site (related to 34 comments). They emphasised the importance of contextualising artefacts to make them more relevant to audiences as part of their everyday reality and ideological viewpoint (e.g., how visitors can personally relate to an artefact displayed in a museum). This is considered crucial to generate opinions and engagements [26].

Making objects more relevant to audiences as part of their everyday reality means that cultural institutions need to reach their audiences outside their spaces. For example, audiences could be reached ahead of a visit as part of an (onsite) experience. Early discovery may also create a better understanding of cultural content once audiences are inside a dedicated cultural heritage space. Streamlining the discovery and experience of cultural heritage could be an avenue for exploration, leading towards an online multimedia museum with connections to the physical world. *PLUGGY* (Pluggable Social Platform for Heritage Awareness and Participation) is a recent project [3] aimed at doing this. *PLUGGY* is a social platform with curatorial tools to allow citizens to create personalised stories and exhibitions online and to share them through social networks with friends, associates, and professionals. In creating these online exhibitions accessible anywhere, users of *PLUGGY* can add their own assets as well as assets from public libraries and museum libraries. Another project that has explored the use of current leading social media in discovering culture beyond museum walls is *PULSE* (2012), an auditory display of twitter data created by McGookin and Brewster to allow users to gain a vibe or understanding of the people, places, and activities that occur in the user's current locale.

Another opportunity we foresee in extending the experience space is by facilitating documentation after events, by sharing this with the public, and by making it easier for visitors to provide feedback or to continue discussions (i.e., related to 11 comments). Findings show that activities are usually documented and shared in close networks of museum friends and, to some extent, using common social media channels such as Facebook and Twitter. Smaller venues such as galleries may not provide follow ups to events at all because of limited resources; factual or visual information about how was an event are either for personal records or for internal purposes and rarely shared with the public. Therefore, the emergence of digital means such as social media allows visitors to start discussions related to their visits in online public spaces, but these discussion may not always reach cultural venues. As a consequence, relevant key players who are responsible for delivering the experience being discussed online remain unaware about these discussions.

In some occasions, however, cultural experiences should not be shared on social medial at all. For example, for intangible cultural heritage such as traditional dances, the actual experience cannot fully be conveyed online and social media can only be a means to share and find events. According to Oscar *"Participation is as close as a 'truthful' experience, where one should travel to where it was created"*. Quin adds that *"People hold different interpretations of a dance which has lead to traditional dances going through a constant transformation from generation to generation"*. Hence, the value of intangible heritage lies in immediate and offline interactions. This should be taken into account when creating digital experiences.

### 3.2.3. Theme 3: Decentralising Collaborations

Thirdly, based on our findings, collaboration should be decentralised and content should be reflected upon by a general audience. Cultural institutions are often not able to provide reflections on user-curated contributions. Once contributions are hosted in institutions online, there is no guidance in making them "useful". According to Alfred, *"when visitors come with objects to his museum there is usually no strategy for evaluating them"*. Hence, we have not reached the goal of "becoming social" that has been set within the new *museology* introduced by Kidd [27].

Allowing a general audience to reflect on cultural content, however, requires a robust and reliable systematic review of who can be considered an expert. A main advantage with this approach is that it would allow for different perceptions on cultural content (linked to 30 transcription items). Our museum professionals acknowledge that they are not necessarily experts in all their collections. To solve this, they involve larger numbers of people to help understanding and curating collections. However, there remains a validation challenge. Moreover, validating the accuracy of audiences' contributions is even a bigger challenge. Different cultures may have conflicting stories, with none being objectively right or wrong. Additionally, there is no standard way or infrastructure to validate them.

Previous research showed that content created by a museum might not be as engaging as content created by others [30]. Involving visitors, locals, or the general audience in interpreting collections and in the validation process of content could in fact lead to a more social structure and, hence, a better understanding and appreciation of objects. Currently, most objects in a museum do not really have that voice. The language in museums prevents a full understanding of the artefact and hence opinions being created.

The collective creation of experiences by users is usually referred to as co-experience [31]. Empowering collaboration in generating content creates opportunities to connect objects and people. Grounding objects better within their original communities and environments is expected to allow visitors to gain "a greater worldview or a new set of connections and correlations between various dimensions of experiences" [32]). This is expected to lead to transformational experiences, which happen if we discard old ways of thinking and provide new opportunities for individuals to invent knowledge, transform what they have encountered in the past, and contribute to new ideas and concepts.

According to Batterbee and Koskinen [33], people design and create experiences collectively on a daily basis. However, co-experience is not about creating products or art; it is about the ways in which participants make things meaningful for each other. One example of co-experience is the *Curarium*, a digital interactive platform created by metaLAB at Harvard. *Curarium* employs crowd sourcing to annotate, curate, and augment works of art within and beyond their respective collections [34]. *Curarium* is an application for exploring, analyzing, and making arguments about the works of art in art collections, and it allows users to annotate these works of art, tell stories about each work, and curate collections in a collaborative way. Similarly, the *Google Art Project* was created in 2011, currently running as Google Arts & Culture platform in collaboration with 151 museums from 40 countries. The online platform enables people to access high-resolution images of more than 32,000 artworks, allowing them to take virtual tours of partner museums galleries with the use of various multimedia features. Users can explore physical and contextual information about artworks and can compile and share their own virtual collections [26].

*3.3. Summary*

Based on our interview results, we identified three themes for opportunities in need of more attention if we want to improve participatory action, accessibility, and diversity in cultural heritage. These include (1) redefining the meaning of "expert" by including also local audiences, (2) extending the experience space beyond cultural heritage sites by allowing offsite discoveries as part of an onsite experience, and (3) providing an infrastructure to decentralise collaboration for cultural heritage. While the first was raised based on issues mentioned by general audience interviewees and relates to Kidd's marketing frame, the other two were based on the main challenges that cultural heritage professionals are currently facing and relate to Kidd's inclusive and collaborative frames [27].

**4. Study 2: A Case Study**

To further support findings from the in-depth interviews, we conducted a case study around built heritage using audio narratives. Heritage has a compelling role in postmodern tourism, specifically towards the built side of things. Architectural collections are popular tourist attractions, which according to Nuryanti can be defined as *the heart of cultural tourism* [35]. The relationship between tourism and built heritage, however, brings to frictions that usually emerge due to the juxtaposition between tradition and modernity. Built heritage can be used as an artefact to promote ethics, history, or industry but can also be experienced by visitors through a narrow understanding of time and place. Architecture can be an immersive space into the history of the city, but the general lack of knowledge towards architecture prevents people from engaging with the built environment. An intriguing perspective is that architecture could act as a common ground that is usually shared by a mixed public, including tourists and locals, while limited information is available to contextualise the artefacts. For this case study, we chose the square at Kensington Gore in central London, which contains such a built heritage, allowing views of historical buildings and monuments. The built heritage of interest in this square are the following:

- the Royal Albert Hall;
- the Prince Albert Sculpture, known as the "Great Exhibition Memorial";
- the Royal College of Music Building;
- the Albert Court; and
- the Beit Quadrangle.

This square is usually a space that both local audiences and tourists share during office breaks or while passing through. See Figure 1 for some views of the site.

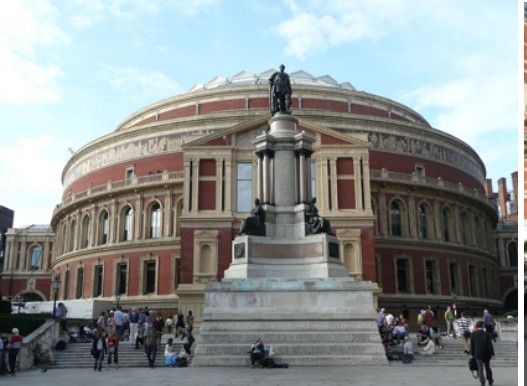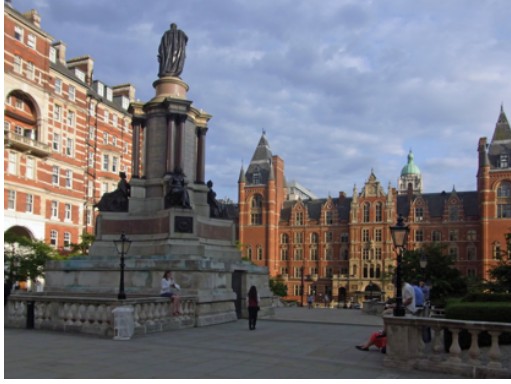

**Figure 1. Left image**: view of the site with the Prince Albert sculpture and Albert Hall in the background. **Right image**: view of the site with the Royal College of Music Building and the Albert Court in the background.

The main objective of this study was twofold: first, to broaden the experience around the buildings and monument with public contribution and, second, to compare engagement between a traditionally/professionally curated audio narrative and a user-generated one.

The results of this study were expected to provide further details on implications on how existing social media could be better utilised to improve user engagement and to encourage distributed curation within a cultural heritage context, looking specifically at the challenges that this approach could generate.

### 4.1. Methodology

#### 4.1.1. Audio Narratives

Firstly, for each built heritage listed in the previous section and shown in Figure 1, two types of audio narratives were created: a traditional curator narrative and a user-generated narrative. The curator narrative was created in partnership with an architectural historian who is trained in London's Victorian urban development. The user-generated narrative consisted of short excerpts inspired from public posts that were created during a co-design workshop. In this workshop, we explained to local audiences the concept of user-generated content. They visited the case study site and were invited to share their thoughts about the site on Instagram. To help them on the topic, we asked them the following questions:

- What does this building remind you of?
- Can you identify any specific artistic or architectural features?

Participants were asked to share their thoughts on Instagram using images and captions and were free to post as many or as little as they wanted. The public posts were then used to extract quotes for the user-generated narrative. Both curator narratives and user-generated narratives were kept short to avoid "museum fatique" [36]. See Appendix B for the transcripts of the narratives.

#### 4.1.2. The 3D Tune-In Toolkit

Once recorded, all narratives were processed using the binaural spatialisation technique through the 3D Tune-in Toolkit [37] in order to create a sense of direction and space within the audio scenes. Binaural spatialisation is a technique that allows for the creation of three-dimensional soundscapes through a simple pair of headphones and is widely used in interactive audio and Virtual Reality (VR) applications. Recorded audio was processed using the 3D Tune-In Toolkit Test application (https://github.com/3DTune-In/3dti_AudioToolkit/releases, accessed on 30 January 2020), creating an immersive soundscape with sound sources located all around the listener at different distances. The spatialised stereo audiofiles were then exported and used for the test.

Recently, within the work of the *PLUGGY* project, a Javascript version of this platform was released called PlugSonic Soundscape, which allows us to perform the same process and to create immersive binaurally spatialised soundscapes using a simple web interface [38].

Immersive virtual environments have been around for several years [39] and have been extensively used in various research fields, both in audiovisual [40,41] and audio-only modes [42]. Previous research has indicated that immersive VR systems can stimulate the experience of spatial presence, albeit this might be mainly based on the visual communication channel [40]. This could be a key outcome considering that, in this study, we are exploring the role that immersive audio could play when looking at making cultural heritage more accessible.

#### 4.1.3. Design Probe

The binaural audio narratives were made available through a simple website created using WordPress. The website included five separate pages for each building or monument; in each page, there was an audio file that could be played back when pressing a specific button on the interface. Each page also had a text version of the narrative. To allow participants to easily access the content, the audio narratives pages were also directly accessible through five QR codes printed and attached to the sides of a laser-cut cube that participants hold on to in their hands throughout the experience (see Figure 2). Each cube also had an image of the building or monument printed and attached to the cube so that

participants could quickly find the audio narrative of interest. By scanning a QR code, participants were immediately directed to the correct web page. On one side of the cube, open questions were presented to trigger participants to comment on the narratives: *"What does this building remind you of?"* and *"What influences do you see?"*. Participants were invited to add their comments or pictures on social platforms such as Instagram using a specific hashtag or in the news and micro blogging platform *Medium* (www.medium.com, accessed on 30 January 2020) using specific tags.

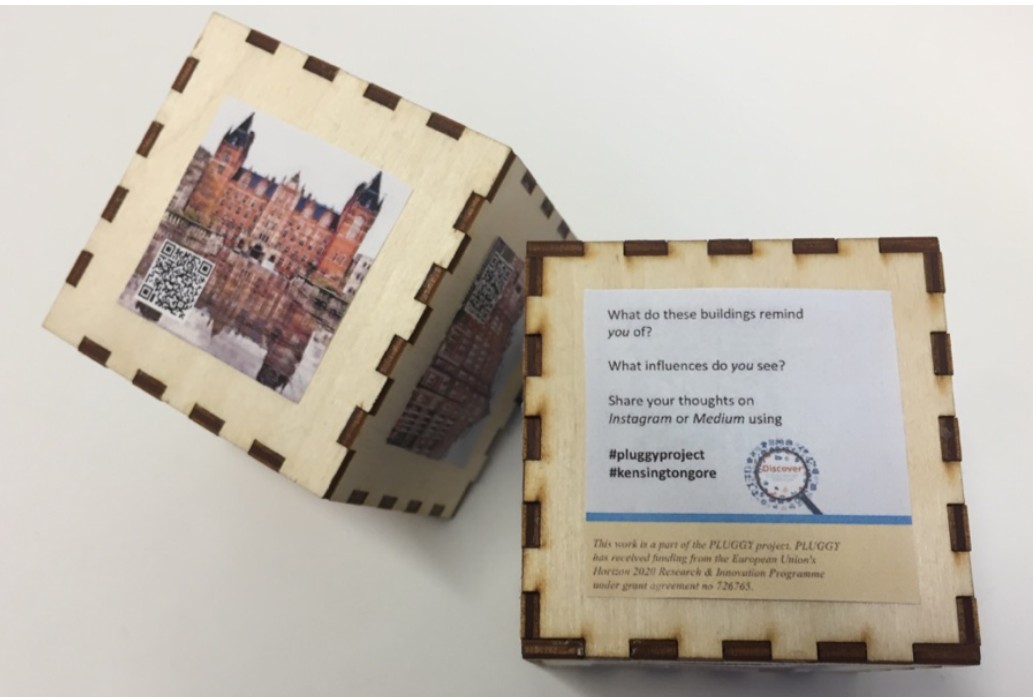

**Figure 2.** Each participant used a cube with images and QR codes on the various sides to access the text and audio information for each target. On one side of the cube, questions were shown to trigger visitors thoughts when roaming around.

### 4.1.4. Participants

Through opportunity sampling, a total of 28 participants were recruited either through the local university network or were randomly approached on the spot while walking through Kensington Gore. Fifteen participants were given access only to the curator narrative. Thirteen participants were given access to both the curator and the user-generated narrative. The study was approved by the Imperial College Ethics commission.

### 4.1.5. Procedures

The study took place in Q2 of 2019. At the start of the study, participants were introduced to the cube and explained how to access the narratives. No specific instructions were given other than to freely select the audio narratives in the order they wanted. They were allowed to walk anywhere within the site. Participants were either given a smartphone and noise cancelling earphones, or they used their own devices to access the audio narratives. Since the site was not close to a main road and has generally not many passing visitors during the week, background noise was not considered an issue [43]. During their explorations, participants were shadowed and observed. A map of the test site was used to annotate navigation, touch points, and behavioural insights. Participants decided when they finished by reporting back to the observer. At the end of study, they were invited to answer a short version of the MEC-SPQ (Measurements, Effects, Conditions Spatial Presence Questionnaire) to measure engagement [44]. The MEC-SPQ was designed for immediate assignment after media exposure, and the short version consists of 28 question items representing seven constructs including process

factors (attention allocation, spatial situation model, self location, and possible actions), variables referring to states and actions (higher cognitive involvement and suspension of disbelief), and variables addressing enduring personality factors (i.e., domain-specific interest). Each construct contains 4 questions that can be answered on a 5-point Likert scale ranging from 1 ("I do not agree at all") to 5 ("I fully agree"). Here, a short description of each constructs follows:

- attention allocation, i.e., to understand which media was more effective in communicating, written text, images, or audio;
- spatial situation model, i.e., to understand the users' ability to describe to what extent it was possible to link the content to their physical surroundings;
- self-location, i.e., to understand how grounded the users felt in relation with the mixed-media environment;
- possible actions, i.e., the sensation of being able to take action in the mediated environment;
- higher cognitive involvement, i.e., to understand how the content activated the thinking of the users;
- suspension of disbelief, i.e., to understand if the users were able to spot errors in the content or were biased toward the user-generated content; and
- domain-specific interest, i.e., to understand how much the users were interested in the topics related to the study.

The group of participants who had the option to listen to both the curator narrative and the user-curated narrative was asked to comment on and to explain their preferences. Demographics were also collected at the end.

### 4.1.6. Analysis

Descriptive statistics were used to analyse the MEC-SPQ scores for participants with access to the curator narrative vs. both curator and user-generated narratives. We also conducted a thematic analysis, similarly to what we did in the interview study but specifically focusing on participants demographics, and their decisions and comments with regard to their preferences. Again, pseudonyms are used to refer to participants.

### 4.2. Results

Overall, the curator narrative was found to be more engaging. Specifically, participants who experienced the curator narrative gave higher scores for self location, possible actions, higher cognitive involvement, and domain-specific interest (see Table 1). The scores differ from participants who experienced the user-generated content by at least 0.5. For the other constructs, no differences were found.

**Table 1.** Average MEC-SPQ (Measurements, Effects, Conditions Spatial Presence Questionnaire) scores for each construct for participants who experienced the curator narrative vs. those who experienced the user-generated narrative. In bold are the scores with at least a difference of 0.5 in comparison to the other participant group.

| MEC Spatial Presence Item Category | Curator Narrative M(SD) | User-Generated Narrative M(SD) |
|---|---|---|
| Attention allocation | 3.77(0.93) | 3.83(0.68) |
| Spatial Situation Model | 3.54(1.18) | 3.60(1.18) |
| Spatial Presence: Self Location | **3.87(1.10)** | 3.32(1.32) |
| Spatial Presence: Possible Actions | **3.79(1.28)** | 3.00(1.17) |
| Higher Cognitive Involvement | **4.24(0.66)** | 3.56(1.03) |
| Suspension of Disbelief | 2.74(1.23) | 2.74(1.16) |
| Domain Specific Interest | **3.77(1.25)** | 3.23(1.11) |

The Role of Familiarity

Based on the qualitative data (i.e., observations and user comments), participants showed preferences for the curator narrative because they trusted it more than the user-generated narrative. This was the case especially for participants who were less familiar with the context and who, therefore, wished to grasp most of the knowledge about the area from an expert. For example, Ella said *"I'm from Sydney, Australia, and I just arrived in the UK yesterday. I'm not very familiar with the surroundings and I'd like to know more about that statue (referring to the Albert Memorial) from a more official source."* On the other hand, participants who were more familiar with the surroundings preferred the user-generated narrative in order to check other people's experience about the same place, i.e., Paul said *"I live in London since six years now and I work nearby since four years. I think, letting people share their experience about the same places is very nice, especially because I'm always interested to get to know what other Londoners know about a specific area."*

The cultural background also seemed to play a key role in participants' preferred narrative type. It made a difference on the amount and the diversity of details the users mentioned to be able to grasp from the surroundings, which is supported by previous research [45]. For example, some participants were not always familiar with the Latin inscriptions under the Albert Memorial monument, and they expected a prompt translation from the audio narrative. This was the case for participants who came from eastern Asia and some European countries where Latin is not taught in secondary school. An audio narrative required a high attention span for non-English native speakers, especially if the content was delivered in a speech form. For this reason, participants in this specific case preferred to also read the text while listening to the audio narrative.

Only when participants were familiar with the heritage site (because of a previous experience related with the site or because of the cultural background) or have a similar cultural background did they prefer to listen to the user-generated narrative. Therefore, in order to consider local audiences as experts or locals as an important marketing channel, the cultural background and knowledge of the content consumers should be taken into consideration. This supports previous research [46]; user-generated content may only be interesting to those with familiarity of the context [47].

*4.3. Summary*

Based on the results of this second part of the study, the curator narrative was generally the preferred one. The only exceptions occurred when participants were already familiar with the site; in that case, the user-generated content was more appreciated. This seems to suggest that curator content should always be presented while user-generated stories can be offered as an additional experience depending on visitors familiarity with the artefact or environment.

## 5. Discussion and Conclusions

This work supports previous projects such as CURIOS [1], among others, that have shown the potential of social platforms to build networks through distributed contributions of users but within the context of cultural heritage. In this work, we identified issues and challenges in involving people in distributed curation in cultural heritage and explored a potential design space and opportunities for social media and audio technology to improve accessibility and awareness and to promote local heritage [23].

Our findings show that current social media could be further utilised to improve distributed curation, accessibility, and diversity in cultural heritage within 3 themes: firstly, that we should consider local audiences as experts and, therefore, as an important resource and potential marketing channel to become more social within the new museology [27]. However, as we have learned from the case study, we do need to take into account visitors' cultural background and level of familiarity. These seem to have some impact on visitors' preferences in curator vs. user-generated content. The latter may only be interesting to those familiar with the site. Hence, the type of content should be tailored not only to

different cultural backgrounds but also to different knowledge levels, which depend on the audience rather than the cultural topic itself. This finding, however, may only be applicable to the type of built heritage used in this case study. We do have to note that this may be different for different types of sites, e.g., pleasant scientific sites vs. dark history sites.

We have also shown the need to extend the experience space by blending the discovery and experiences of heritage artefacts into everyday spaces. In the case study, we faced a major challenge in engaging visitors who were passing through. We believe the challenge may have been in the nature of the context as its part of participants' "everyday" or "work" environment and visitors were just passing through (i.e., a transitional place) e.g., to get to one of the museums nearby. More exploration should be performed on how cultural heritage can be introduced into people's "everyday" or "transitional" spaces, where relevant heritage cstories could be discovered that are related to other heritage sites or artefacts inside a museum nearby.

Additionally, there seems to be a need for an infrastructure to decentralise collaboration for cultural heritage, so that general audiences can be content creators anywhere and at anytime for objects onsite and/or offsite museums or heritage sites. In order to allow a general international audience to create content, technology should support easy access to curation tools that can be used by anyone and at anytime. The rising importance of digital distributed curation is important to improve inclusion and may even become more valued during current times such as during the COVID-19 pandemic [4,48].

Our work contains a few limitations. Although we highlighted a number of opportunities for further exploration, our findings do not offer a complete insight into potential opportunities in social media to better engage people at a global level. This is due to the qualitative nature of our studies. Generally, the sample size used in qualitative research methods is smaller than in quantitative research methods because qualitative research methods intend to gain in-depth understanding of a phenomenon and, as in this work, are often centered on the how and why of a particular issue. As described in [49], in-depth interview work is not as concerned with making generalisations to a larger population of interest and does not tend to rely on hypothesis testing but rather is more inductive and emergent in its process. Therefore, we selected interviewees based on expected types of initial users of PLUGGY. Another limitation is that, in addition to motivating visitors to participate on the spot, they may need to help support quality checks of distributed curation. This was not addressed in this work. Furthermore, it is also important to note that, when encountering foreign cultures, each has different ways to approach content. As we have seen, not only do peoples' cultures but also familiarity and previous experiences play significant role. A meaningful cultural experience, therefore, needs to be presented in different ways for different audience types.

To conclude, the work presented here could be of value in helping to identify opportunities in the use and design of social media and technology, which could further support participatory action and, hence, inclusivity. Inclusivity is a key to promote awareness. Additionally, wthout proper access to cultural content, people are unable to engage in a meaningful experience with cultural heritage. However, further work is necessary to explore different case studies within various types of transitional locations and media content types.

**Author Contributions:** Conceptualization, V.L., S.K. and L.P.; methodology, V.L. and S.K.; software, L.P.; validation, V.L. and S.K.; formal analysis, V.L. and S.K.; investigation, V.L. and S.K.; resources, V.L. and L.P.; data curation, V.L., S.K. and L.P.; writing—original draft preparation, V.L. and S.K.; writing—review and editing, V.L. and L.P.; visualization, V.L. and S.K.; supervision, L.P.; project administration, L.P.; funding acquisition, L.P. All authors have read and agreed to the published version of the manuscript.

**Funding:** *PLUGGY* received funding from the European Union's Horizon 2020 Research & Innovation Programme under grant agreement no 726765. The content reflects only the author's view, and the European Commission is not responsible for any use that may be made of the information it contains.

**Institutional Review Board Statement:** The study was conducted according to the guidelines of the Declaration of Helsinki, and approved by the Imperial College Research Ethics Committee (reference: 17IC4197; approved in October 2017).

**Informed Consent Statement:** Written informed consent was obtained from the patient(s) to publish this paper.

**Acknowledgments:** We greatly thank our project partners Luca Simeone, Spyros Souvlakis, Luis Molina-Tanco, and Silvia Brandelesi for helping with the transcriptions and providing feedback on initial drafts of the paper. This work is a part of the *PLUGGY* project.

**Conflicts of Interest:** The authors declare no conflict of interest. The funders had no role in the design of the study; in the collection, analyses, or interpretation of data; in the writing of the manuscript; or in the decision to publish the results.

## Appendix A

*Appendix A.1. Questions Asked to Cultural Heritage Professionals*

1. Tell me a bit about yourself. What you do on a daily basis?
2. What projects have you been working on or will be working on?
3. What is the process of content creation up to delivery to general audience?
4. What was the role of social media?
5. What were the main goals to achieve?
6. What were your main challenges?
7. What it is that made something (e.g., an aspect of technology) successful or not?
8. What do your audience want and like(consider different segmentation of audience)?
9. Have you observed any shifts of use or behaviours among your audience?
10. What would you like to do in the future that you cannot do right now?
11. How do you think about crowdsourced content? What challenges do you foresee?

*Appendix A.2. Questions Asked to a General Audience*

1. Tell me a bit about yourself. What do you do on a daily basis?
2. Can you tell me about cultural heritage in your home city?
3. Have you been to any cultural heritage related sites or events recently? Tell me about it?
4. What did you do in preparation of the visit?
5. What were your main motivations and goal of the visit?
6. When and where did you get information?

## Appendix B

*Appendix B.1. Curator Narratives*

Appendix B.1.1. Royal Albert Hall

*"It's beautiful isn't it? The Royal Albert Hall. It seems like only yesterday when I was talking to Henry Cole, chairman of the royal society of arts, about my dream to build a permanent exhibition hall. A sort of 'follow up' of the Great Exhibition of 1851. Oh, by the way, my name is Albert... and I was the Prince of Saxe-Coburg and Gotha. I had a beautiful wife, Victoria, queen of England. If it wasn't for her and Henry, the royal albert hall wouldn't exist. The Hall was opened on 29 March 1871 by Queen Victoria, renamed in my memory to the Royal Albert Hall of Arts and Sciences. When Victoria opened the Hall, she was so overcome by emotion that the Prince of Wales had to speak in her place... She was one of the most beautiful women on this world. Unfortunately, I wasn't there at the inauguration. But they told me that as she laid the foundation stone, she said 'It is my wish that this hall should bear his name to whom it will have owed its existence and be called The Royal Albert Hall of Arts and Sciences'. But the Royal Albert Hall is part of my dream resulted in the construction of Albertopolis. The area is here in South Kensington and contains several structures dedicated to the arts and sciences. For example, I'm now facing the Imperial College London. Ah. the Imperial College... the name me and Victoria chose for the building was*

*Imperial Institute! Unfortunately, I didn't have time to see the opening of this building either...
due to my early death. The college was one of the first in Britain to teach by experiments rather than
just by lectures."*

### Appendix B.1.2. Royal College of Music

*"Within sight of the Royal Albert Hall are two buildings dedicated to music. The first one, with
your back to the Hall, is the Royal College of Music. It originated from my proposals for a national
music training scheme for young people. Founded in 1882, it is now part of the University of London.
The Royal College of Music is one of the world's leading conservatoires. It provides specialised
musical education and professional training at the highest level for performers, conductors and
composers. Some of the world's best classical musicians have studied here. Listen carefully and you
may be able to hear students playing inside."*

### Appendix B.1.3. Albert Court

*"The Albert Court is one of the first historical Flat housing facilities in the UK. This mansion
block was originally conceived by the freeholders and commissioners, as the first stage of a larger
private development. Building began in 1890 to the designs of Frederick Hemings but, following the
collapse of the Liberator Building Society in 1892 and the death of Hemings in 1894, the building
had only reached the 3rd floor level. The Albert Court was finished by R.J. Worley's designs between
1896 and 1900. The time it took to complete this building reminds me the troubled and long path
that the Royal Albert Hall had to go in order to become the magic building the whole world envies to
Britain. This building is made with red brick with elaborated stonework bands and dressings. It has
6 storeys, a D-shaped plan, with the North and East facades following the crescent line of the former
Royal Horticultural Society's Garden."*

### Appendix B.1.4. Beit Hall

*"Beit Hall, otherwise known as the Beit Quadrangle, is one of Imperial College London's oldest
and most historic buildings. It was funded by Sir Otto Beit in 1910, then director of the governing
body of Imperial College. Initially it was a faculty building housing Biology and later converted
to a student Hall. The band Queen is said to have performed here its first gig. Their music was
definitely different to what I was used in the 19th Century. During term-time, it is one of the largest
Halls of Imperial College providing self-catering accommodation to 340 students. But during the
remaining 14 weeks Beit magically transforms into an international conference centre and hotel.
The colour that architect Aston Webb chose for the bricks perfectly matches the orange of my beloved
royal albert hall."*

### Appendix B.1.5. Albert Memorial Statue

*"Look at me! This statue, which represents me, is a memorial to the Great Exhibition. Until
1891 it stood in the garden of the Royal Horticultural Society, which used to be on the site of
the Royal College of Music. Take a look at the inscription below this statue, which sets out how
much money we raised from the Great Exhibition. Albertopolis actually all started with the Great
Exhibition in 1851, which showcased Britain's international role in the arts and sciences. It was
the first of a series of world fairs for culture and industry, where scientists used to meet, discuss
and share ideas and innovations. From the Exhibition's success this area south of Hyde Park was
established as a long-term legacy to celebrate science, technology, culture and the arts. We have
seen how Albertopolis is home to some of the world's leading museums, academic institutions and
national organisations. Each one of these is continually evolving and expanding, through their
buildings and the people who work in, study in and visit them. More than 150 years later Prince
Albert and Sir Henry Cole's legacy is still alive and well. Albertopolis is still at the heart of the arts
and sciences."*

*Appendix B.2. User-Generated Audio Narrative*

Appendix B.2.1. Royal Albert Hall:

Excerpt 1: *"These decorations you see under the roof remind me of pottery from ancient Greece"*—sound location: top right.

Excerpt 2: *"The reliefs remind me of the Greek reliefs in the British Museum taken from the Parthenon. They are definitely different but somehow I like the idea of relief representation that it comes from the Greek influence?"*—sound location: top left.

Excerpt 3: *"I used to dream of being a musician. Whenever I see the building, I wonder what it would have been like to be a musician, and I feel a bit of pity for my lost dream."*—sound location: middle front.

Excerpt 4: *"The architecture of the façade is surprisingly similar to the San Felice Sul Panaro church near Modena, which sadly has been destroyed in the recent earthquake."*—sound location: middle front.

Appendix B.2.2. Royal College of Music:

Excerpt 1: *"On both side of the main entrance there are musical themed reliefs, and in the spandrels above them are carved wreath motifs."*

Excerpt 2: *"You can see the pilasters travel up the building and change into two small niches, which then move upwards and change into finials on the central dormer. This has a clock mounted in the top section with a triangular sun relief and strap work decorations."*

Appendix B.2.3. Albert Memorial Statue:

Excerpt 1: *"I'm curious about the 4 statues placed in the base of the Albert memorial. Their attire looks different from each other, maybe they symbolise the 4 corners of the world?"*

Excerpt 2: *"In the other Albert Memorial in Hyde park there are some Marble figures and animals representing Europe, Asia, Africa and America at each corner. For the animals, Africa is represented by a camel, the Americas with a buffalo, Asia by an elephant and a bull for Europe."*

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
