# Peer review of "Towards a More Accessible Cultural Heritage: Challenges and Opportunities in Contextualisation Using 3D Sound Narratives"

_applsci, doi:10.3390/app11083336_

Round 1
Reviewer 1 Report
The work concerns sound narratives in the context of an 'enhanced' experience of cultural heritage. In section 2, three themes were discussed, based on interviews. In section 3 audio narratives of a case study were recorded and proposed to test participants through binaural processing.
The work is a part of the PLUGGY project, and I tried to read more about that.
Even if all in the paper is interesting, the text should be summarised.
For instance, some previous projects cited in the Background section were already mentioned in the "Design and Evaluation of a Web- and Mobile-Based Binaural Audio Platform for Cultural Heritage", published on the same special Issue. It is intentional?
Some references are missing (253)
Moreover, I'm wondering whether the authors took into account two points concerning audio narrative:
1) the duration of each content / the duration of the whole experience. They could be related to the so-called 'museum fatigue'
https://www.tandfonline.com/doi/full/10.1080/10645570903203406
2) the background-noise due to human activity
https://www.sciencedirect.com/science/article/pii/S0360132320305485
On the one hand, this kind of noise can influence the intelligibility of speech. On the other hand, human activity - both the soft or loud ones - can influence the soundscape around the visitor.
Even if not treated in this research - these two points deserve to be mentioned in the context of 'audio narrative', from the reviewer's point of view.
Author Response
# 1 - Even if all in the paper is interesting, the text should be summarised. For instance, some previous projects cited in the Background section were already mentioned in the "Design and Evaluation of a Web- and Mobile-Based Binaural Audio Platform for Cultural Heritage", published on the same special Issue. It is intentional? Comunita's paper is part but a continuation of the same project, PLUGGY. Therefore, it is intentional that both papers refer to the same project. However, the background and introduction in Comunita's paper is focusing on different key topics. # 2 - Some references are missing (253) We have added the missing references. # 3 - Moreover, I'm wondering whether the authors took into account two points concerning audio narrative: 1) the duration of each content / the duration of the whole experience. They could be related to the so-called 'museum fatigue' https://www.tandfonline.com/doi/full/10.1080/10645570903203406 2) the background-noise due to human activity https://www.sciencedirect.com/science/article/pii/S0360132320305485 On the one hand, this kind of noise can influence the intelligibility of speech. On the other hand, human activity - both the soft or loud ones - can influence the soundscape around the visitor. Even if not treated in this research - these two points deserve to be mentioned in the context of 'audio narrative', from the reviewer's point of view. Following reviewers suggestion, we have included a reference to Bitgood in section 3.1.1 with regard to the length of the narratives and we included a reference to D’Orazio in section 3.1.5 with regard to the background noise.
Reviewer 2 Report
The present manuscript addresses design and presentation potentials for modern data sets in the context of cultural heritage, with a specific concentration on 3D sound narratives. The study involves findings derived from 22 interviews.
The topic is timely, and this paper is dedicated to an interesting case study on 3D sound narratives. The manuscript has the potential to be published. Before the final acceptance, the authors should consider several aspects in a revision of the manuscript:
- Could you please extend your introduction and emphasize the relevance why potential design opportunities have to be explored when dealing with participatory approaches in cultural heritage? This key aspect is missing in the first chapter (which, by the way, should be chapter 1 and not 0).
- The manuscript lacks references when introducing immersive (binaural) soundscapes (in VR) in chapter 3.1.2. There is some existing literature that deal with the potentials of 3D sound in immersive virtual environments. These papers are worth considering:
Edler, D., Kühne, O., Keil, J., Dickmann, F. (2019c): Audiovisual cartography: established and new multimedia approaches to represent soundscapes. In: KN – Journal of Cartography and Geographic Information, 69 (1), pp. 5-17.
https://doi.org/10.1007/s42489-019-00004-4
Hruby, F. (2019): The sound of being there – audiovisual cartography with immersive virtual environments. In: KN – Journal of Cartography and Geographic Information, 69 (1), pp. 19-28. https://doi.org/10.1007/s42489-019-00003-5
Moreover, authors should think of citing Murray Schafer’s pioneering monography when dealing with soundscapes:
Schafer, R.M. (1977): The soundscape. Our sonic environment and the tuning of the world. Rochester.
- The topic of sound narratives could also consider other reported study that combine (audio-visual) storytelling with mobile devices, such as this study published by Indans and colleagues:
Indans, R., Hauthal, E., Burghardt, D. (2019). Towards and audio-locative application for immersive storytelling. KN – Journal of Cartography and Geographic Information, 69(1), pp. 41-50. https://doi.org/10.1007/s42489-019-00007-1
- The discussion and conclusion section (4) rather reads like a summary of the interview results. However, these results are not connected to the similar research from the state-of-the-art literature in this field. Could authors please show the broader message of this study in order to make a clear contribution within the ongoing debates?
Author Response
- # 1 - Could you please extend your introduction and emphasise the
- relevance why potential design opportunities have to be explored when dealing with participatory approaches in cultural heritage? This key aspect is missing in the first chapter (which, by the way, should be chapter 1 and not 0).
Section number is changed to 1. We have adapted the intro to clarify the aim earlier in the paper.
# 2 - The manuscript lacks references when introducing immersive (binaural) soundscapes (in VR) in chapter 3.1.2. There is some existing literature that deal with the potentials of 3D sound in immersive virtual environments.
We have added paragraphs to better introduce binaural soundscapes in section 4.1.2.
- # 3 - The discussion and conclusion section (4) rather reads like a summary of the interview results. However, these results are not connected to the similar research from the state-of-the-art literature in this field. Could authors please show the broader message of this study in order to make a clear contribution within the ongoing debates?
As per reviewer suggestions, we hope to have clarified this better in the discussion by adding new paragraphs.
Reviewer 3 Report
The text refers to an interesting and timely issue and as such merits publication. It would however benefit from a thorough review, taking into account several issues:
Introduction
In the introductory section of the text – “setting the scene” for it – the authors could emphasise more what is new and unique in their research, what research gaps do they intend to fill in? The issue of research gaps is introduced very vaguely as late in the text as p. 6.
What is “participatory action in cultural heritage” – does it only refer to crowdsourcing? The broader picture could be painted (e.g. including crowdfunding and relevant literature).
How does participatory approach to narrating heritage relate to the broader issues of the uses of heritage to develop/strengthen social capital? It would be beneficial to frame the article more strongly in this vein of heritage literature, in particular if as one of the conclusions the authors point to the greater /promising aspects with respect to local audiences/residents for whom sharing non-professional narratives on heritage could indeed help to feel more connected to each other and their place of residence (ref. numerous publications in the past decade and the role of cultural institutions to enhance social capital and creativity (e.g. museums, libraries, heritage interpretation centres) (numerous publications in the past decade ranging from Murzyn-Kupisz and Dzialek 2013 to Bell Wallace 2020). An interesting reference with respect to digitalisation of heritage would also be Borowiecki et al. 2016.
The issue of diverse audiences – literature on heritage stakeholders should also be taken into account (e.g. Howard 2003)
The authors refer to Design Thinking – there is ample literature on that – should be referred to.
How to their findings reflect on the changing roles of heritage professionals with respect to heritage narration? Another idea in terms of deeper rooting of the text in heritage literature would be to make reference to the authorised and non-authorised heritage discourse of Smith and the idea of heritage as a process of meaning making.
Methodology, in particular selection of respondents
The authors should explain how the interview participants were selected – justify how/in what way the number of interviews is sufficient, have they ensured that there was a sufficient diversity in terms of geographical scope of countries/types of heritage professions/heritage institutions/heritage types represented in the first study (no information on this issue was provided in the text)? How were the members of the general audience recruited? General audience is a very broad term – how was it ensured that they represent diverse socio-demographic backgrounds, types of heritage audiences in terms of frequency of heritage visits, gender, age, education, cultural background, etc.?
This issue is important both for the first case study and in particular for the second case study where the participants’ place of residence and cultural background seem to be important for the conclusions from the study (hence: what cultural background, age, education level, gender, place of residence – local/London/England/foreign did the participants have?)
On p. 12 line 554 the authors mentioned that demographics were collected at the end (at the end of what?/again how were the participants recruited?, was proper sample selection ensured?)
p. 7 onward – from time to time the authors refer/quote particular interviewees – interviews should be ordered/coded/given numbers -using some coherent method/
How was the information from interviews analysed? Using what methods?
There is also no information on when the case studies took place (at least year and month or time-span)
Other comments
p. 8 359 the authors mention that “smaller venues” do not do follow ups on events – while this may be true – such places should not be scorned for not doing as much as their more prominent/major counterparts – large heritage institutions have more resources (including human resources) to engage in time consuming and often technologically complex follow up activities – this is a limitation of this research as well and should be acknowledged (this actually makes smaller institutions underprivileged and more likely to be excluded from this digital revolution).
From a more general perspective, perhaps the authors could consider if putting two case studies in one text is really the best use for their data – maybe it would be better to write to separate texts but take more time to analyse the case study results – reflect on them in more depth. Perhaps instead of (in my opinion redundant) presentation of exact description provided to study participants for case study two (Appendix B) the discussion of results of both case studies could be elaborated to a greater extent. Last but not list the impact of COVID-19 pandemic could be mentioned on the limitations of the study and the rising (?) importance of digital narrations of heritage.
The authors could also reflect more on when user generated content is useful and when the curator narrative is particularly important (e.g. a relatively pleasant/”easy” heritage site such as the one referred to in London versus a dark heritage site).
Minor and technical issues
I am not a native speaker but it seems to me that the article would benefit from very careful proofreading
Some awkward/unclear phrases:
p. 4 “people participatory action in cultural heritage” – what is that
p.7 /line 298 “Dialogue in social media often come from a small number …” come or comes…
p. 7 line 315 “when visitors are in the moment with no plan” – awkward/what do the authors mean?
p.7 line 318 “creating space on their phone” – I understand what the authors refer to but in a scientific article it should be explained in a less colloquial manner
p.7 line 340-1 what is “a discovery stage” – it is unclear as other stages are not explicitly mentioned
p. 12 line 544 an awkward phrase, extend or extent?
p. 14 line 633 “Further work should focus… “a very awkward sentence – hard to understand what the intended thought was
p. 17 reference 4 – why are capital letters used for the publication title?
Author Response
# 1 - In the introductory section of the text – “setting the scene” for it – the authors could emphasise more what is new and unique in their research, what research gaps do they intend to fill in? The issue of research gaps is introduced very vaguely as late in the text as p. 6.
We made updates in the introduction to better clarify the problem and aim of the work.
# 2 - What is “participatory action in cultural heritage” – does it only refer to crowdsourcing? The broader picture could be painted (e.g. including crowdfunding and relevant literature).
We changed it to distributed curation or user-generated content to stay consistent with the rest of the paper.
# 3 - How does participatory approach to narrating heritage relate to the broader issues of the uses of heritage to develop/strengthen social capital? It would be beneficial to frame the article more strongly in this vein of heritage literature, in particular if as one of the conclusions the authors point to the greater /promising aspects with respect to local audiences/residents for whom sharing non-professional narratives on heritage could indeed help to feel more connected to each other and their place of residence (ref. numerous publications in the past decade and the role of cultural institutions to enhance social capital and creativity (e.g. museums, libraries, heritage interpretation centres) (numerous publications in the past decade ranging from Murzyn-Kupisz and Dzialek 2013 to Bell Wallace 2020). An interesting reference with respect to digitalisation of heritage would also be Borowiecki et al. 2016.
We referred to Murzyn-Kupisz and Dzialek 2013 in section 2.1. Unfortunately we were not able to find Bell Wallace 2020. Please provide the doi or link to the article.
# 4 - The issue of diverse audiences – literature on heritage stakeholders should also be taken into account (e.g. Howard 2003)
Unfortunately, we were not able to find it. Please provide the doi or link to the article.
# 5 - The authors refer to Design Thinking – there is ample literature on that – should be referred to.
As suggested, we have added the reference.
# 6 - How do their findings reflect on the changing roles of heritage professionals with respect to heritage narration? Another idea in terms of deeper rooting of the text in heritage literature would be to make reference to the authorised and non-authorised heritage discourse of Smith and the idea of heritage as a process of meaning making.
Unfortunately, we were not able to find it. Please provide the doi or link to the article.
# 7 - The authors should explain how the interview participants were selected – justify how/in what way the number of interviews is sufficient, have they ensured that there was a sufficient diversity in terms of geographical scope of countries/types of heritage professions/heritage institutions/heritage types represented in the first study (no information on this issue was provided in the text)? How were the members of the general audience recruited? General audience is a very broad term – how was it ensured that they represent diverse socio-demographic backgrounds, types of heritage audiences in terms of frequency of heritage visits, gender, age, education, cultural background, etc.? This issue is important both for the first case study and in particular for the second case study where the participants’ place of residence and cultural background seem to be important for the conclusions from the study (hence: what cultural background, age, education level, gender, place of residence – local/London/England/foreign did the participants have?)
We clarified recruitment in sections 3.1.1 and in 4.1.4. Generally, to cover a more complete scope of countries and different types of heritage professionals could have resulted in hundreds of interviews. Instead we tried to focus only on potential early adopters as our target group, which we added as a limitation in our discussion section.
# 8 - On p. 12 line 554 the authors mentioned that demographics were collected at the end (at the end of what?/again how were the participants recruited?, was proper sample selection ensured?)
We clarified the method of recruitment in 3.1.1 and in 4.1.4.
# 9 - p. 7 onward – from time to time the authors refer/quote particular interviewees – interviews should be ordered/coded/given numbers -using some coherent method
We have given pseudonyms to our interviewees and hope to have clarified method in sections 3.1.3. and 4.1.6.
# 10 - How was the information from interviews analysed? Using what methods?
We used the MAXQDA Tool for tagging and categorising data and then grouped themes. For the first study this is described in section 3.1.3. For the second study, in 4.1.6., we refer to the same approach as mentioned in 3.1.3, to avoid redundancy.
# 11 - There is also no information on when the case studies took place (at least year and month or time-span)
We added details on when it took place in sections 3.1.2. and 4.1.5.
# 12 - p. 8 359 the authors mention that “smaller venues” do not do follow ups on events – while this may be true – such places should not be scorned for not doing as much as their more prominent/major counterparts – large heritage institutions have more resources (including human resources) to engage in time consuming and often technologically complex follow up activities – this is a limitation of this research as well and should be acknowledged (this actually makes smaller institutions underprivileged and more likely to be excluded from this digital revolution).
We sure agree with that. For this reason, we spoke with professionals from museums as well as professionals working in galleries. Our overall intention with PLUGGY is to provide a tool that is also accessible to these smaller venues since they have limited resources.
# 13 - From a more general perspective, perhaps the authors could consider if putting two case studies in one text is really the best use for their data – maybe it would be better to write to separate texts but take more time to analyse the case study results – reflect on them in more depth. Perhaps instead of (in my opinion redundant) presentation of exact description provided to study participants for case study two (Appendix B) the discussion of results of both case studies could be elaborated to a greater extent. Last but not list the impact of COVID-19 pandemic could be mentioned on the limitations of the study and the rising (?) importance of digital narrations of heritage.
The studies were conducted pre-covid but we added a reference in the discussion section.
# 15 - The authors could also reflect more on when user generated content is useful and when the curator narrative is particularly important (e.g. a relatively pleasant/”easy” heritage site such as the one referred to in London versus a dark heritage site).
From our findings and approach, we are unaware in which other context UGC is useful. No other types of cultural site have been evaluated in this study. Therefore, instead we reflect on this limitation in the discussion.
Minor and technical issues
# 17 - Some awkward/unclear phrases:
p. 4 “people participatory action in cultural heritage” – what is that Corrected
p.7 /line 298 “Dialogue in social media often come from a small number …” come or comes… Corrected
p. 7 line 315 “when visitors are in the moment with no plan” – awkward/what do the authors mean? Corrected
p.7 line 318 “creating space on their phone” – I understand what the authors refer to but in a scientific article it should be explained in a less colloquial manner Corrected
p.7 line 340-1 what is “a discovery stage” – it is unclear as other stages are not explicitly mentioned Corrected
p. 12 line 544 an awkward phrase, extend or extent? Corrected
p. 14 line 633 “Further work should focus… “a very awkward sentence – hard to understand what the intended thought was Deleted as it is redundant with previous sentence.
p. 17 reference 4 – why are capital letters used for the publication title? Corrected
Round 2
Reviewer 2 Report
The authors prepared a revised manuscript version in which they considered the points of review round no. 1. The response letter is rather short and does not lead to detailed answers to the points. However, the currenct manuscript version is acceptable for publication, in my opinion.